# Movement Compensated Driver's Respiratory Rate Extraction

Young-Keun Yoo and Hyun-Chool Shin *

Department of Electronic Engineering, Soongsil University, Seoul 06978, Korea; erric02033@soongsil.ac.kr
* Correspondence: shinhc@ssu.ac.kr; Tel.: +82-828-7165

**Abstract:** In non-contact vital sign monitoring using radar, radar signal distorted by the surrounding unspecified factors is unsuitable for monitoring vital signs. In order to monitor vital signs accurately, it is essential to compensate for distortion of radar signals caused by surrounding environmental factors. In this paper, we propose a driver vital signal compensation method in driving situations, including the driver's movements using a frequency-modulated continuous-wave (FMCW) radar. Driver's movement is quantified from the radar signal and used to set a distortion signal compensation index to compensate for the signal distortion induced in the driving situation that the driver's movement occurs. The experimental results show that the respiration rate estimated from the radar signal compensated through the proposed method is similar to the actual respiration rate than from the signal before calibration. These results confirm the possibility of using the proposed method in a non-statistic situation and effectiveness in estimating respiration rate reflecting human movement in monitoring vital signs using FMCW radar.

**Keywords:** FMCW radar; contactless driver monitoring; vital; respiration; movement; distortion; compensation

## 1. Introduction

As driving a vehicle becomes a major part of our daily life, the incidence of traffic accidents increases due to numerous factors. Driver breathing abnormalities, such as apnea or hyperventilation, and drowsy driving are among the main causes of traffic accidents every year. According to a recent study, approximately 20% of traffic accidents are caused by drowsiness, which accounts for a large proportion of all accidents [1,2]. Detecting driver breathing abnormalities and drowsy driving in advance and implementing real-time measurement can thus prevent major accidents. A highly accurate driver vital monitoring study becomes necessary in this regard [3,4].

Vital monitoring methods for conventional drivers were conducted by cameras and motion sensors inside the vehicle to monitor factors that are external to the body, such as the driver's eyelid closing [5] and nodding of the head or using electromyography (EMG) and electrocardiogram (ECG) sensors [6,7]. However, conventional methods have difficulties in directly recognizing the driver's movements in low-illuminance environments, such as night driving. In addition, methods using EMG and ECG sensors require physical contact between the driver's skin and the sensor [8,9] or have problems such as the use of the internal vehicle space for sensor installation.

Driver vital sign monitoring using frequency-modulated continuous-wave (FMCW) radar [10] is a non-contact method that uses electromagnetic waves and has the advantages of low power consumption and small packaging [11], solving the problems of conventional methods. Conventional methods of monitoring driver vital signals [12–14] estimate the respiration rate in situations where the driver's condition is stable and the driver's movement is not incorporated. However, in the actual driving environment, monitoring vital signs accurately is very difficult because of external factors such as the driver's movements while driving [10]. The FMCW radar is a distance-based communication device that uses electromagnetic waves. The driver's movement causes irregular fluctuations in the radar

signal [15], which reduces the accuracy of vital signal monitoring through the FMCW radar [16]. Therefore, it is necessary to compensate for the respiration signal reflecting the movement of the driver in the movement situation to improve the accuracy of signal extraction [17,18].

In this paper, we introduce a new signal compensation index setting method that detects the driver's movement using a 60 GHz bandwidth FMCW radar and compensates for the distorted driver's respiration signal. To verify the accuracy of the proposed algorithm, we monitored the driver's vital signals for seven driving situations, depending on the driver's movement. We quantified the driver's movement and used it to set a signal compensation index. In the results, respiration rates estimated from the driver's respiration signal, which was compensated by the proposed method and that without compensation, were compared with the actual driver respiration rate, in contrast to the conventional vital signal monitoring method.

## 2. Methods

### 2.1. FMCW Radar

Radar is a detection device that transmits electromagnetic waves and modulates the received signal reflected from the target. Radars are divided into several types according to the modulation method of the transmitted and received signals. This study considers FMCW radar using component modulation in terms of frequency. The FMCW radar transmits a chirp signal with a linearly increasing frequency. The difference in frequency between the transmitted and received signals is defined as an intermediate frequency (IF). After mixing the transmitted and received signals of the FMCW radar, a low-band filtering process is performed to extract the IF signal [19].

$$x(t) = M(t) \cdot cos(2\pi(S \cdot t_d)t + (2\pi \cdot f_{carrier} \cdot t_d - \pi \cdot S \cdot t_d^2)) \tag{1}$$

$S$ is the linear frequency increase in the chirp signal defined by $\frac{BW}{T_c}$, $BW$ is the bandwidth of the FMCW radar, $T_c$ is the chirp duration, $t_d$ is the time delay between the transmitted and received signal, $f_{carrier}$ is the carrier frequency, and $M$ is the magnitude of the radar signal. Moreover, the transmitted and received signals are processed within microseconds. Therefore, because $t_d \gg t_d^2$ , (1) is approximated as (2).

$$x(t) = M(t) \cdot cos(2\pi(S \cdot t_d)t + (2\pi \cdot f_{carrier} \cdot t_d)) \tag{2}$$

The FMCW radar estimates the distance to the target using the frequency and phase of the IF signal. If the distance between the radar and object changes, the time delay between the transmitted and received signals is converted into the phase component $R(t)$ by the change in target position, where $R(t)$ is $\frac{2(\varnothing(t) - \varnothing(t-1))}{c}$. The IF signal is remodeled as (3); $\varnothing(t)$ is the phase of the current signal.

$$x(t, n) = \sum_n M(t, n) \cdot cos(2\pi \left( \frac{S \cdot 2d}{c} \right) n + (2\pi \cdot f_{carrier} \cdot R(t))) \tag{3}$$

$n$ is the sampling index of the chirp, and $d$ is the distance from the target. Fast Fourier transform processing was performed to extract the frequency and phase components of the IF signal.

$$X(t, k) = \sum_{n=1}^{N-1} x(t, n) \cdot e^{\frac{j \cdot 2\pi \cdot k \cdot n}{N}} \tag{4}$$

$N$ is the number of samples in the chirp, and $k$ is the range resolution index of the radar. Vital signals, such as respiration or heartbeat, are detected using the phase variation of the intermediate signal over time.

$$\varnothing(t) = \sum_{k_{first}}^{k_{end}} \angle\, X(t,k) \tag{5}$$

where $\left[ k_{first}\ k_{end} \right]$ is the target range in which an object is assumed to exist.

### 2.2. Vital Signal Detection

Figure 1 shows an example of a driver's respiration signal measured by the FMCW radar. Figure 1a is an image of the driver's IF signal, and the color bar shows the amplitude of the IF signal. Time is the scan time, n is the sampling time in a chirp. The section where the driver movement occurred is indicated at the top of the figure. The difference in the displacement of the signal caused by the driver's breathing forms a wave pattern, which indicates respiration. Irregular distortion of the respiration signal occurs in the movement section. However, in the other sections, we can see that the driver's respiration signal is regular. For comparison with the reference respiration, a respiration signal extracted from the radar is derived using the phase component of the signal obtained after fast Fourier transform processing of the IF signal. Phase component was derived from the radar signal of a specific range bin. The specific range refers to the distance between the radar and the driver and was extracted using the magnitude-phase coherency (MPC) [20]. The reference respiration signal was acquired by a pressure sensor worn on the abdomen. Figure 1b shows a comparison between the respiration derived from the IF signal and that from the reference sensor. Figure 1b demonstrates that if movement occurs, the radar signal is distorted compared to the reference respiration. Thus, to accurately detect the driver's breathing, it is necessary to compensate for the respiration signal in situations where the driver is not stationary.

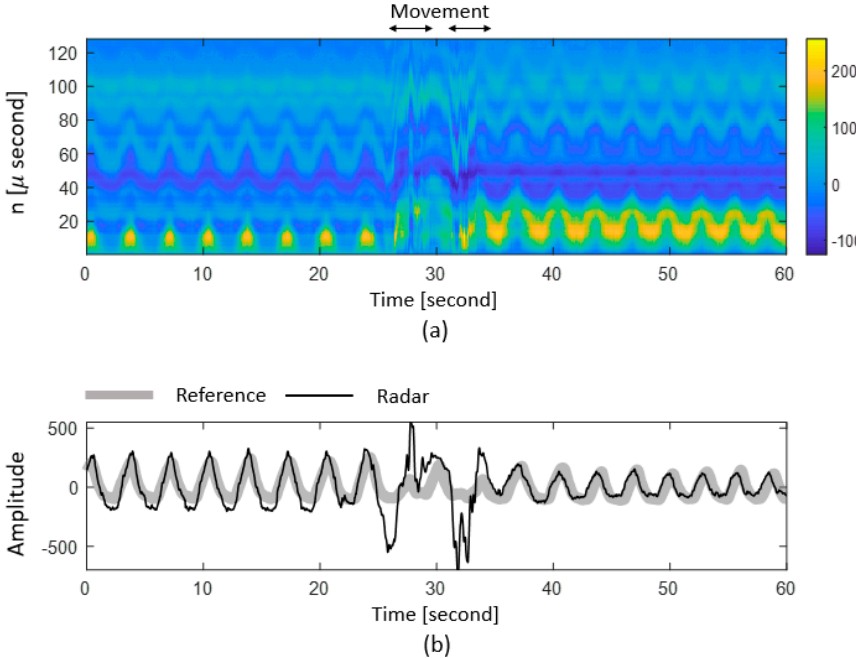

**Figure 1.** Measured respiratory signals. (**a**) Radar signal, (**b**) respiratory signal measured by radar at a specific range bin and from actual breathing.

### 2.3. Movement Quantification

In this study, we assumed that the respiration signal in the movement section was a distorted signal. To compensate for the radar signal irregularly distorted by motion [21], we defined the difference in the signal magnitude as a motion index $\mu(t)$. $\mu(t)$ is derived by averaging the difference in magnitude from the signal immediately preceding in time for the specific range [22].

$$\mu(t) = \frac{1}{k} \sum_{k} |X(t,k) - X(t-1,k)| \tag{6}$$

Figure 2 depicts the accuracy of the motion index for driver motion detection. To ensure that $\mu(t)$ detects the driver's movement accurately, we compared it with the actual movement of the driver, which was detected by an acceleration sensor. Figure 2a shows the driver's respiration signal and the driver's motion index value derived from the driving situation where there is movement. The radar signal was measured by installing a radar inside the driver's car seat. For the reliability of the proposed motion index, Figure 2b shows the driver's body acceleration value measured by the acceleration sensors, which were worn on the driver's chest, right wrist, and foot. From the variability of the measured acceleration values, it is possible to check the section in which the driver moves within the measurement time. In the section where driver movement occurred, as shown at the top of Figure 2a, irregular distortion of the driver's respiration signal occurred. Not only the driver's chest, which is the transmission direction of the radar signal but also the movement of the arms and legs can cause a fine displacement difference in the driver's body, resulting in distortion of the signal. Therefore, compared with Figure 2b, it can be seen that distortion of the respiration signal is caused not only by the driver's body but also by the movement of the arms and legs necessary for driving. We confirmed that the motion quantification index reflects the driver's movement by matching the section where the driver's movement occurs with the fluctuation of the motion quantification index.

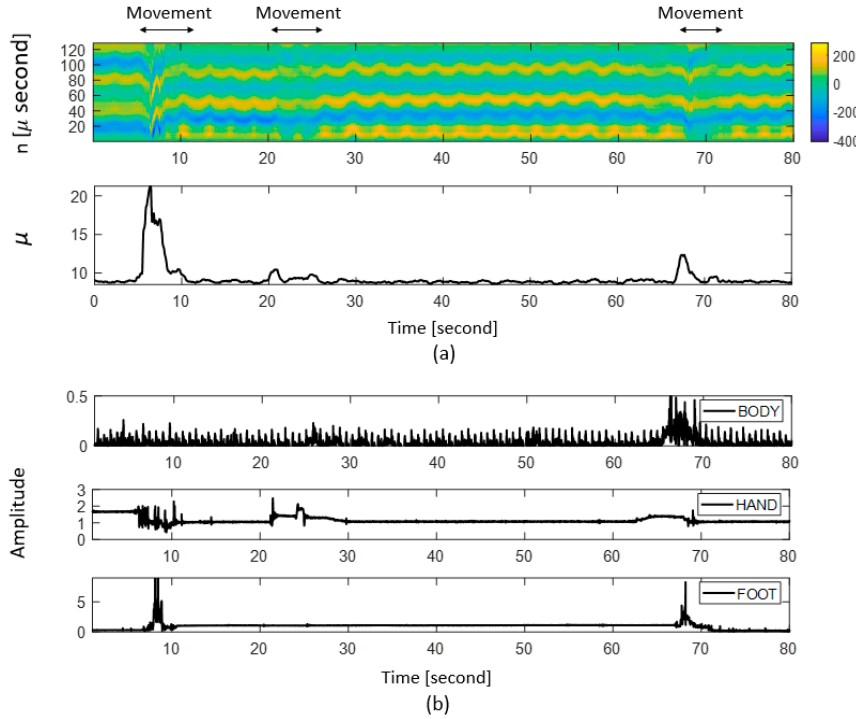

**Figure 2.** Comparison of the quantification accuracy of motion index. (**a**) Respiratory signal and motion quantification index, (**b**) acceleration value for different parts of the driver's body (chest, back of the hand, and back of the foot).

### 2.4. Distortion Compensation

We introduce a method of compensating a distorted signal using a motion quantification index to detect respiration more accurately than the conventional detection method in a movement situation. Figure 3 shows the distortion signal compensation index that sets the signal compensation interval. $f\mu(t)$ converges to 0 when the driver's movement quantification index increases above the threshold and to 1 when it decreases [23]. The threshold value is set as the average of the movement quantification index during the entire measurement time.

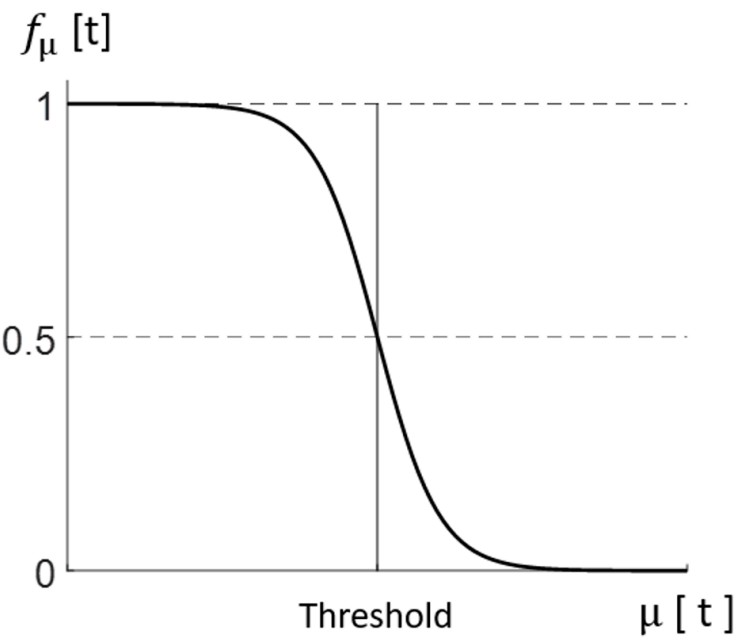

**Figure 3.** Signal compensation index reflecting motion quantification index.

$f\mu(t)$ is expressed by using (7). The signal compensation sector was set according to the degree of motion, quantified from the motion index.

$$f\mu(t) = \frac{e^{-\alpha(\mu(t)-threshold)}}{1 + e^{-\alpha(\mu(t)-threshold)}} \tag{7}$$

We devise the compensation of signal distortion as (8) using $f\mu(t)$.

$$\hat{\varnothing}(t) = f\mu(t)\cdot\varnothing(t) + (1 - f\mu(t))\cdot\hat{\varnothing}(t-1)) \tag{8}$$

$\varnothing(t)$ is the phase of the fast Fourier transform processed IF signal. If $\mu(t)$ decreases below the threshold, $f\mu(t)$ converges to 1, reflecting the phase value of the current signal as the final phase. Conversely, if $\mu(t)$ increases beyond the threshold, $f\mu(t)$ converges to 0, reflecting the immediately preceding value of the final compensated phase to the current phase.

We applied (8) to the driver's respiration signal to check the effectiveness of compensating the distorted signal in the movement section. To compensate for the distortion in a radar signal, we compared the difference in the signal magnitude with time shift. The shift time is set to 1 s. Figure 4 shows the driver signal compensation process distorted by movement and the corresponding variation in the frequency components. The result of the compensated signal due to the driver's movement using the proposed method is shown in Figure 4a. By the movement, $f\mu(t)$ converges to 0, and the phase component at this time is compensated through (8); thus, it can be confirmed that the driver respiration signal is consequentially compensated. Figure 4b shows the frequency component at one point in time when the driver is not in a dynamic state to confirm the accuracy of the

signal compensation. The frequency extracted from the calibrated driver's respiration signal matches that of the actual respiration signal compared with the frequency of the uncompensated signal. Therefore, Figure 4b ensures that the distorted driver respiration signal is compensated by the movement through frequency component comparison.

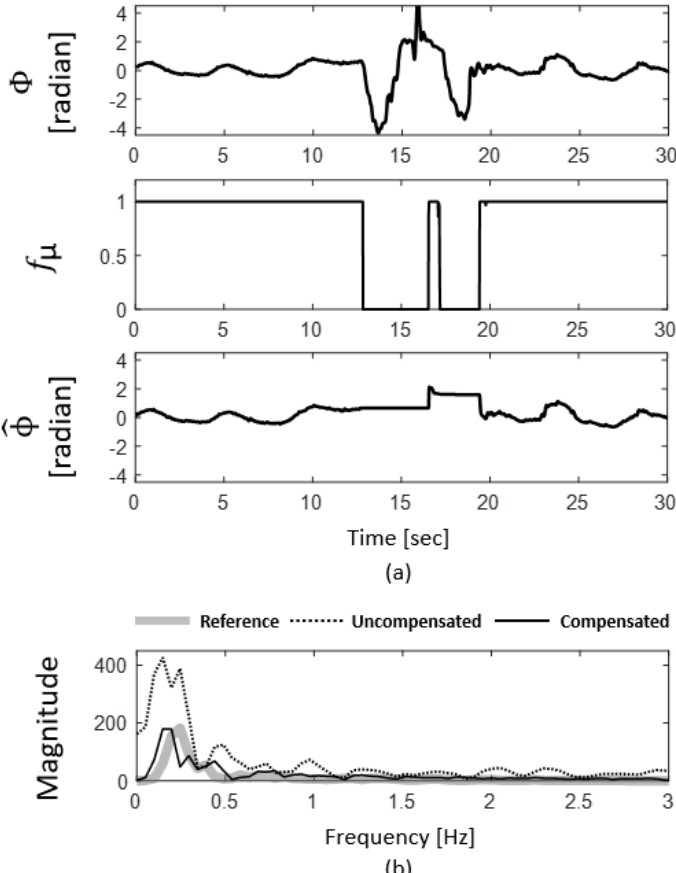

**Figure 4.** Components comparison of respiration signal based on compensation. (**a**) Compensation of distorted phase components of respiration signal, (**b**) comparison of frequency components of respiration signal according to the distortion compensation.

### 2.5. Respiration Rate Estimation

We calculated the driver's respiration rate per minute from the signal compensated using a previously suggested method (8). Figure 5 shows the estimation results of the respiration rate according to the compensation. The respiration rate was estimated by peak tracking [24] using the frequency component of the driver's vital signals. Figure 5a shows the frequencies of the original and distorted respiration signals. The estimated respiratory rate, including the movement situation, is inaccurate compared to the reference respiration rate, as shown in Figure 5c. Contrary to the previous results, Figure 5b represents the frequency of the respiratory signal compensated by the method proposed in (8). The frequency value of the compensated respiration signal is strongly confirmed at 0.35 to 0.4 Hz and is similar to the original signal.

To verify the accuracy of the signal compensation, the compensated respiration rate using the peak tracking technique was compared with the actual respiration rate. As a result, Figure 5d shows that the trends of the compensated original respiratory rate coincide.

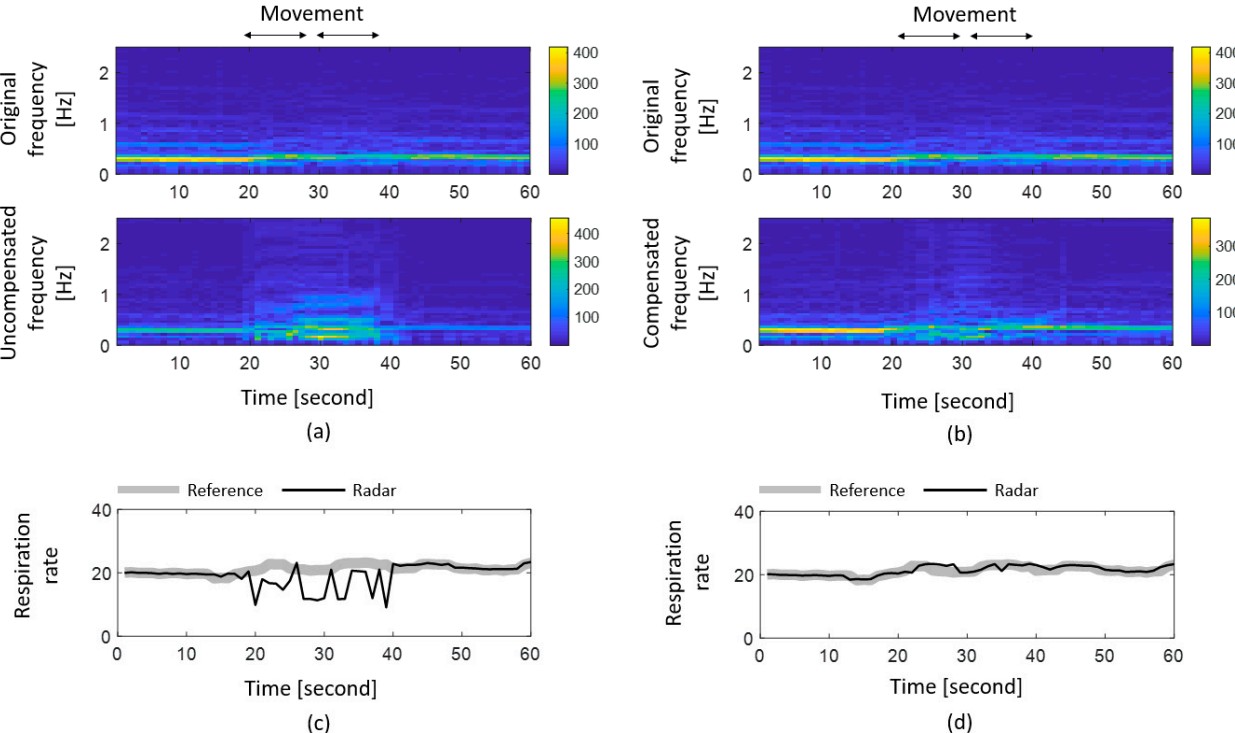

**Figure 5.** Compensation results comparison. (**a**,**b**) Frequency components of driver's respiration signal, (**c**,**d**) estimated respiratory rate.

## 3. Processing

### 3.1. Radar Processing

We conducted all the experimental processes and implemented the proposed algorithm using the FMCW radar (Bitsensing INC., Seoul, Korea) [25] specification in Figure 6 with its parameters used for recording. The radar with these specifications can detect objects that are in front of it within the range of 0–3.187 m. Figure 7 shows the experimental environment for detecting driver vital signals, equipped with an actual vehicle car seat and driving equipment to simulate actual driving conditions. A reference respiration rate for comparison was measured using a Neulog Respiration Monitor Belt logger sensor (Neulog Inc., Rishon-Lezion, Israel). Figure 7a shows the location of the acceleration sensor attachment. The driver's movements were measured using Perception Neuron Studio (Noitom Inc., Beijing, China). We used the acceleration values worn on the driver's chest, right wrist, and foot of the 14 motion sensors attached to the body as circled in Figure 7a. The radar was placed inside the car seat, facing the center of the chest of the driver shown in Figure 7b.

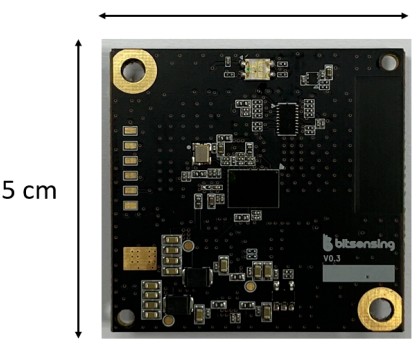

| Parameter | Value |
|---|---|
| Center frequency | 60 GHz |
| Chirp duration | 128 $\mu s$ |
| Sampling frequency | 2 MHz |
| Scan interval | 50 ms |
| Bandwidth | 6 GHz |
| Tx antenna | 1 channel |
| Rx antenna | 1 channel |

**Figure 6.** Single-channel FMCW radar.

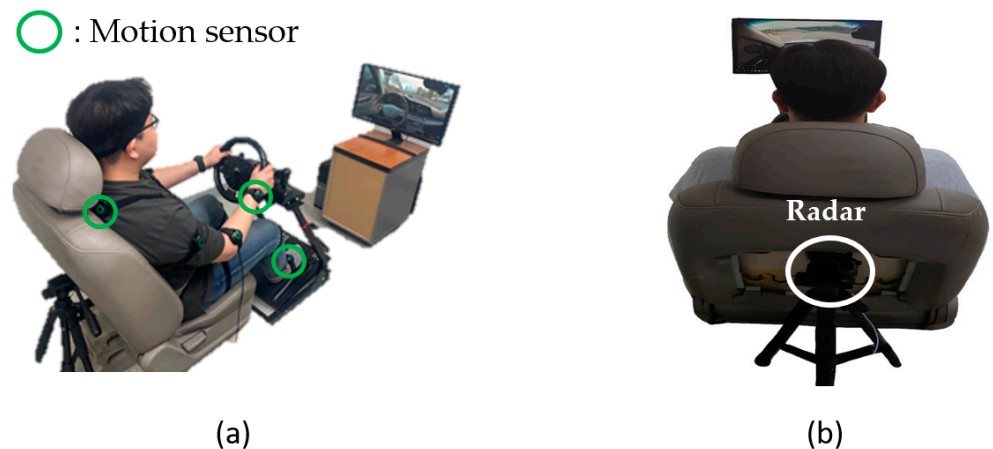

: Motion sensor

(a)                                              (b)

**Figure 7.** Respiratory signal measurement environment. (**a**) Experimental setup and location of wearable motion sensors attachment, (**b**) location of FMCW radar.

*3.2. Experimental Processing*

We classified the driver's vital signal measurement process into seven situations according to the driver's movements in Table 1. Each experiment consisted of one minute, and it was conducted while watching driving videos for each situation on the front display in Figure 7a. In addition, all the drivers were well acquainted with each experimental environment.

**Table 1.** Driving experiment conditions according to the movement.

| Driving Situation | Driver Movement | | Detail |
| --- | --- | --- | --- |
| | O | X | |
| Straight 1 | | √ | Normal breathing for 1 min |
| Straight 2 | | √ | Normal and fast breathing per 30 s |
| Left Turn | √ | | 1. High driver motion value 2. Long movement duration |
| Right Turn | √ | | 1. High driver motion value 2. Long movement duration |
| Complex Driving 1 | √ | √ | Stopping and driving straight |
| Complex Driving 2 | √ | √ | Sudden stopping and lane changing |
| Lane Change | √ | | 1. Low driver motion value 2. Short movement duration |

Because of the experimental characteristics associated with the radar installed inside the vehicle car seat, we limited the object detection range of the radar to 5–27.5 cm. To reduce the signal processing complexity, we selected candidates for the target range from the driver's respiration signal using MPC. Using peak tracking in the frequency domain, we calculated the respiratory rate from the driver's reference and estimated respiration signals. To ensure accuracy of the estimated respiration rate from the compensated respiration signal in the motion situation, we compared it with the actual respiration rate, which was obtained using a respiration sensor.

**4. Result**

*4.1. Respiration Rate Estimation Result*

We compensated the respiration signal for each driving situation according to the occurrence of movement and examined the accuracy of the estimated respiratory rate from the compensated signal. Figure 8 shows the driver's respiration signal and respiratory rate estimation results for a driving situation in which the driver was motionless. Figure 8a,b are both for motionless cases, so the respiration signal is regular, and the corresponding signal compensation index value is 1. As shown in Figure 8c,d, we confirmed that the

respiratory rate estimated from the frequency component of the respiration signal using peak tracking in both driving situations matches the reference respiratory rate.

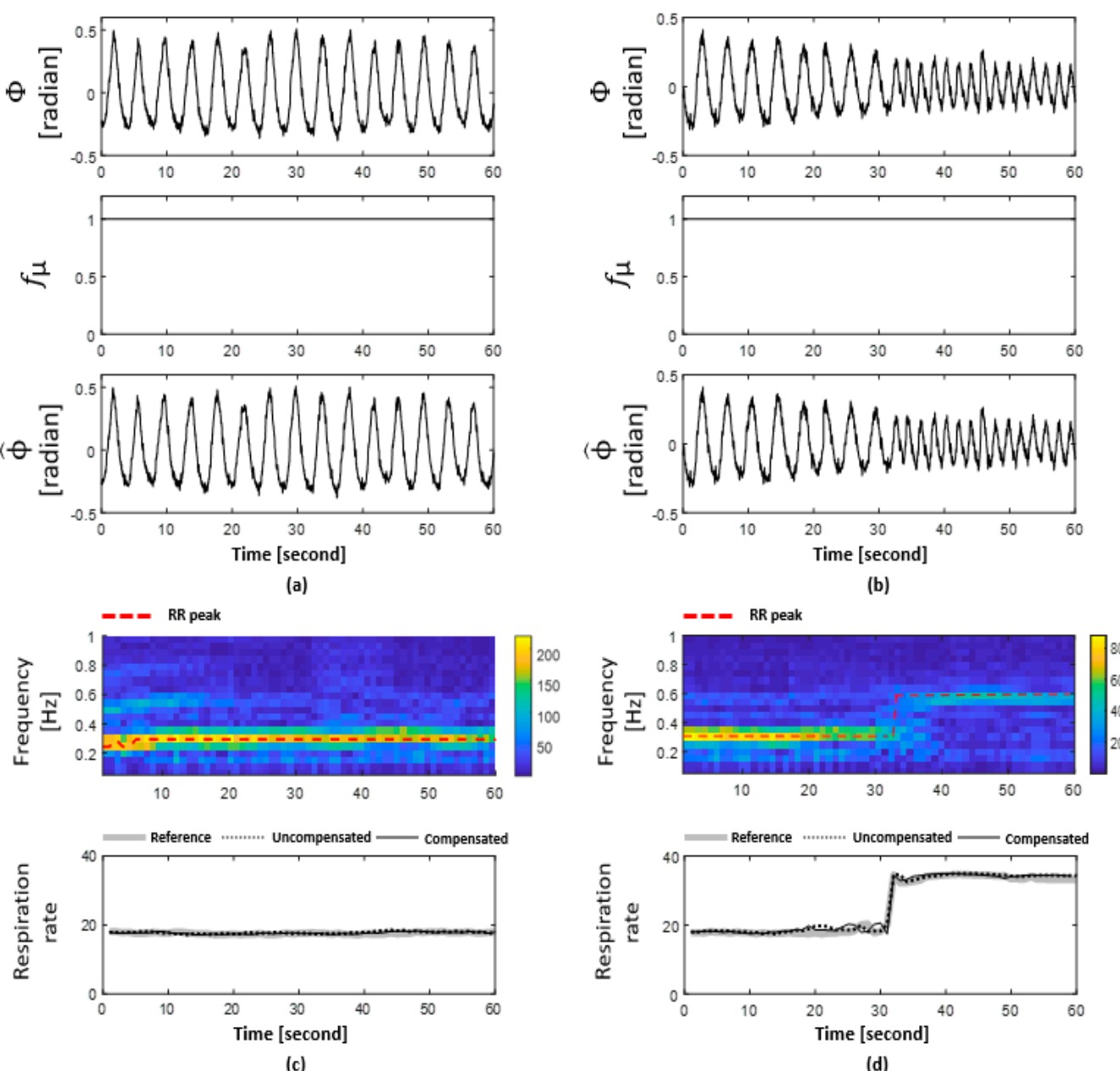

**Figure 8.** Compensated respiration estimation and comparison in situations where the driver is motionless. (**a**,**c**) Respiration compensation in straight driving, (**b**,**d**) respiration compensation in straight driving with faster breathing after 30 s.

Figure 9 shows the results of estimating the driver's respiration signal and respiratory rate in a driving situation accompanied by the driver's movement. Figure 9a–d are the results of the direction change, namely left turn and right turn, respectively. An irregular distortion of the driver's respiration signal occurs in the direction change situation, and it is compensated by (8). The respiratory rate from the compensated driver respiration signal correlates highly with the reference respiratory rate.

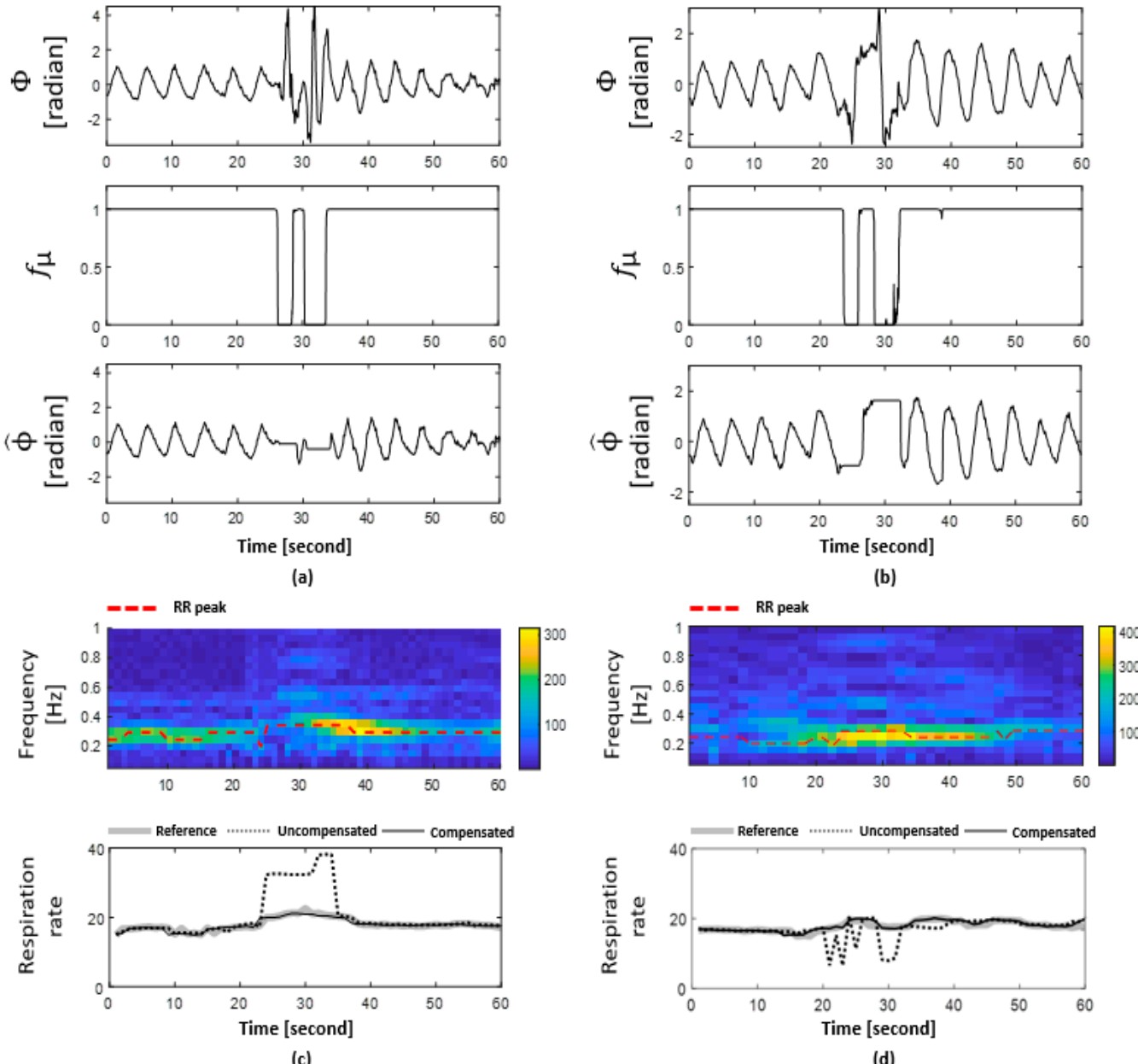

**Figure 9.** Compensated respiration rate estimation and comparison in each situation where the driver's movement is present. (**a**,**c**) Respiration compensation in turn left, (**b**,**d**) respiration compensation in turn right.

Figure 10 shows the results of estimating the driver's respiration signal and respiration rate in driving situations that include lane change and complex movements such as deceleration, acceleration, and straight driving. As direction changes, the signal is distorted in the section where movement occurs. As shown in Figure 10a, a sudden stop is accompanied by a large movement of the driver's body. Thus, the variability in $\mu(t)$ is higher than that in other driving situations. In the case of Figure 10b, after stopping while driving in a straight line, driving is resumed, and $\mu(t)$ is relatively small because the driver's movement is less than that of a sudden stop. In both driving situations mentioned above, the distorted driver signal is compensated by detecting the section where the driver moves. In addition, Figure 10d,e show that the accuracy of the respiratory rate estimation increased through the proposed method. Figure 10c shows that the respiration signal when changing lanes is similar to that of the direction change situation but is associated with

relatively less movement of the driver's body. The distortion of the respiration signal occurs due to body movement when the driver changes lanes. However, there is little change in the driver's body when changing directions, which causes a relatively low change in the motion index $\mu(t)$.

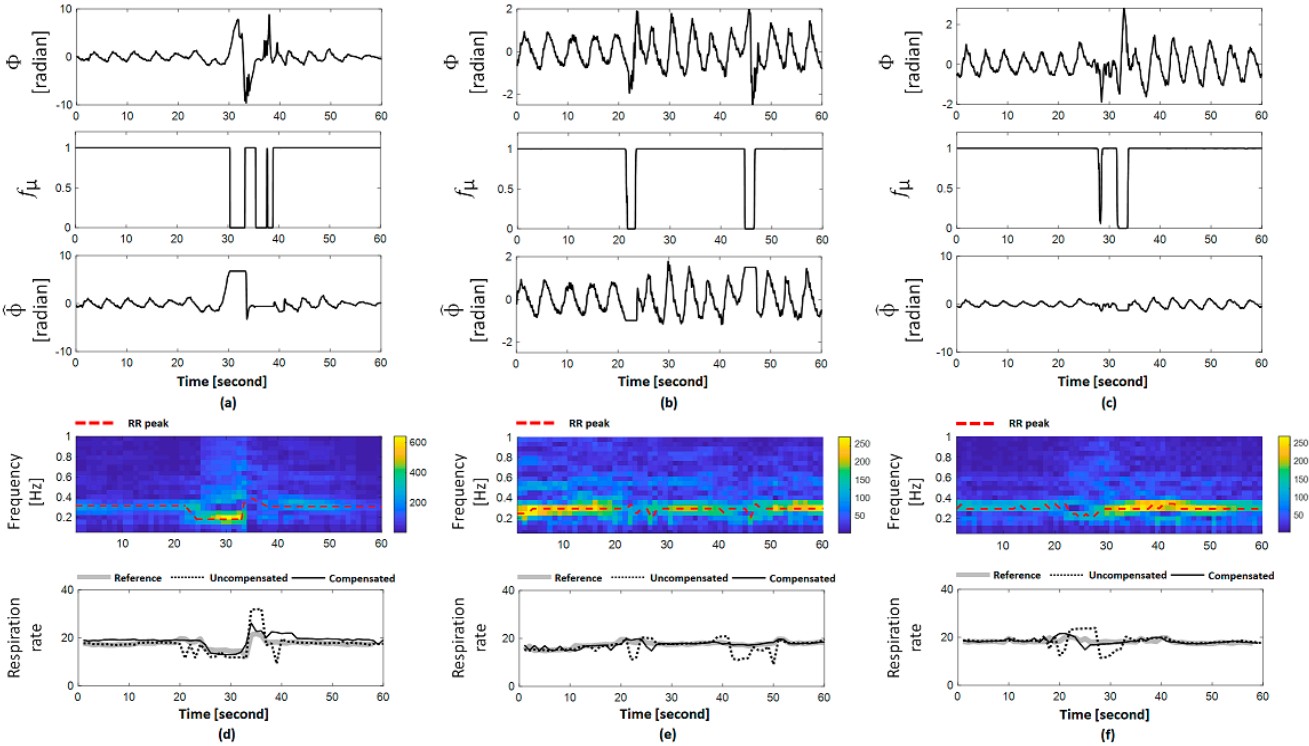

**Figure 10.** Compensated respiration rate estimation and comparison for each situation in which the driver's movement is present. (**a**,**d**) Respiration compensation in sudden stop, (**b**,**e**) respiration compensation in stop and start, (**c**,**f**) respiration compensation in lane change.

We detected the driver's movement when changing lanes and compensated for signal distortion. This movement situation also results in high accuracy of the compensated respiratory rate shown in Figure 10f. The results demonstrate that the proposed method compensates for an irregularly distorted signal in a driving situation in which movement exists, and it accurately estimates the respiratory rate.

### 4.2. Accuracy Analysis

We calculated the mean error and accuracy of the estimated respiratory rate from the compensated respiration signal according to the driving situation for the three drivers. Accuracy is the absolute difference between the reference respiratory rate and the estimated respiration signals. In addition, the error was estimated by the standard deviation of the respiration rate. Table 2 shows the accuracy of the respiratory rate of the drivers for each driving situation. Distortion of the driver's respiration signal only occurs when the driver's movement is detected. Thus, we showed the accuracy of the estimated respiratory rate only for the situation in which movement occurred. The respiratory rate estimated from the compensated respiration signal has a lower error compared to the reference respiratory rate in the case of direction change (left and right turn) and complex driving where a large movement is detected in the driver's body. There are individual differences in body movements for each driving situation. Thus, when changing lanes, the driver's motion quantification value is generally low, and there is almost no signal distortion.

**Table 2.** Respiratory rate accuracy analysis according to each driving situation.

| Driving Situation | Subject # | Respiration Rate Error | | p-Value(ANOVA) |
|---|---|---|---|---|
| | | Before Compensation | After Compensation | |
| Left Turn | 1 | 2.778 ± 5.201 | 0.374 ± 0.489 | 0.0003 |
| | 2 | 1.415 ± 1.542 | 1.077 ± 1.292 | |
| | 3 | 1.210 ± 1.588 | 1.091 ± 1.380 | |
| | Mean value ± SD | 1.801 ± 2.777 | 0.847 ± 0.987 | |
| Right Turn | 1 | 2.097 ± 1.686 | 1.369 ± 1.282 | 0.0106 |
| | 2 | 1.234 ± 1.767 | 0.936 ± 1.492 | |
| | 3 | 2.134 ± 2.095 | 1.488 ± 1.267 | |
| | Mean value ± SD | 1.822 ± 1.849 | 1.264 ± 1.347 | |
| Complex Driving 1 | 1 | 1.541 ± 2.645 | 1.062 ± 1.338 | 0.0372 |
| | 2 | 1.888 ± 1.933 | 1.526 ± 1.698 | |
| | 3 | 3.654 ± 2.921 | 2.961 ± 2.026 | |
| | Mean value ± SD | 2.361 ± 2.500 | 1.850 ± 1.687 | |
| Complex Driving 2 | 1 | 1.260 ± 1.940 | 0.546 ± 0.753 | 0.0075 |
| | 2 | 1.426 ± 2.468 | 1.185 ± 1.935 | |
| | 3 | 1.240 ± 1.947 | 0.694 ± 0.843 | |
| | Mean value ± SD | 1.309 ± 2.118 | 0.808 ± 1.177 | |
| Lane Change | 1 | 2.536 ± 2.118 | 1.012 ± 1.047 | 0.0022 |
| | 2 | 0.884 ± 1.849 | 0.485 ± 0.779 | |
| | 3 | 0.587 ± 0.973 | 0.522 ± 0.657 | |
| | Mean value ± SD | 1.336 ± 1.647 | 0.673 ± 0.828 | |

Therefore, the estimated respiratory rate error is the same for the two subjects, before and after signal compensation. Overall, for most drivers, the proposed method for estimating a driver's respiratory rate has higher accuracy than conventional methods.

## 5. Conclusions

Accurate driver vital signal monitoring during driver movements is a very useful technology for the prevention of major driving accidents caused by respiratory abnormalities. For limited driving situations in an experimental environment, we acquired driver vital signals using a highly portable FMCW radar, which can be packaged in small sizes. Because the accuracy of driver vital signal detection and respiration rate estimation varies greatly depending on the driver's movement, a distorted signal compensation index based on motion quantification is newly introduced for more accurate respiration rate estimation in movement situations. Using the experimental results, we proved that the driver's respiration signal distorted by movement can be accurately compensated using the proposed signal compensation method. The results showed that the respiration rate estimated by the proposed method had improved accuracy compared to that of conventional methods. The proposed signal compensation method is useful for monitoring vital signs in all fields where movement exists during monitoring.

The limitation of this work is that it has been performed offline. To make it be performed in real time, radar needs to be installed in a vehicle, and all the signal processing modules also need to be embedded in the radar sensor module. More experiments and sensitive analysis of the threshold values in real road under driving situations remain for future works for rigorous demonstration.

**Author Contributions:** Conceptualization, Y.-K.Y.; Data curation, Y.-K.Y.; Formal analysis, Y.-K.Y.; Investigation, Y.-K.Y.; Methodology, Y.-K.Y.; Project administration, H.-C.S.; Resources, H.-C.S.; Software, Y.-K.Y.; Supervision, H.-C.S.; Validation, H.-C.S.; Visualization, Y.-K.Y. and H.-C.S.; Writing—original draft, Y.-K.Y.; Writing—review & editing, H.-C.S. All authors have read and agreed to the published version of the manuscript.

**Funding:** This work was supported by institute of Information & communications Technology Planning & Evaluation (IITP) grant funded by the Korea government (MSIT) (No. 2021-0-00305).

**Institutional Review Board Statement:** Not applicable.

**Informed Consent Statement:** Informed consent was obtained from all subjects involved in the study.

**Data Availability Statement:** Not applicable.

**Conflicts of Interest:** The authors declare no conflict of interest.

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
