# Peer review of "Movement Compensated Driver’s Respiratory Rate Extraction"

_applsci, doi:10.3390/app12052695_

Round 1
Reviewer 1 Report
My comments have been addressed.
Author Response
Thank you.
Reviewer 2 Report
- The author should spend more effort discussing the existing prior art of motion compensation for respiratory rate monitoring with Radar (instead of just citing some of the existing work)
- The author still does not explain clearly the meaning/significance of each figure used in the paper. For example, what is the physical parameter/unit of the color bar in figure 1(a) and what is the unit for y axis in figure 1 (b)
- Also from the line 103, author does not explain how are some parameters chosen, for example, how is the "specific range bin " detected?
- equation (7) and (8) suggest that the motion compensation performance heavily depends on the value threshold, please provide more information on how is the threshold calculated/set
- Many figures seem to have low resolution (figure 6,7,8 for example).
- Author should spend more efforts discussing the limitations and potential improvements to the method proposed in this study
- The English writing and presentation still need to be improved.
Reviewer 3 Report
The paper proposes a radar signal distortion compensation method in the driving situation. The article tries to improve the accuracy of estimating respiration rate reflecting human movement in monitoring vital signs using FMCW radar. The paper shows experimental results for identifying the respiration rate for various driving conditions.
I consider that the paper is well structured and provides a clear experimental demonstration for verification of the proposed method.
Here are my comments regarding further improvement.
1. It is unclear whether the work has been done in real-time or off-line. If real-time, please indicate a more detailed experimental setup description. If off-line, please add some anticipated technological requirements to make it real-time. I consider an off-line vital sign assessment would be less valued.
2. In my opinion, the manuscript has limited scientific contribution. Overall, it lacks innovative points except for the distortion index. Therefore, I recommend summarizing your scientific contributions. Further, I also suggest improving the mathematical explanation of your approach to compensate for the signal distortion. Finally, a sensitivity analysis of the threshold values (maybe using a simulation approach) would be excellent.
3. Finally, the justification of using the radar sensor over other non-contact approaches such as camera-based methods could be helpful. Also, a more rigorous statistical hypothesis test such as AVOVA will convince the paper's quality of scientifically meaningful contents.
Author Response
Please see the attachment.

This manuscript is a resubmission of an earlier submission. The following is a list of the peer review reports and author responses from that submission.
Round 1
Reviewer 1 Report
This article proposes a movement compensated driver’s respiratory rate extraction method in this article.
There are several suggestions that may help to improve the article:
- It would be better to add latest related works in the introduction part, especially about motion compensation in radar signal to position the work of this article such as “J. Pan, Z. -K. Ni, C. Shi, Z. Zheng, S. Ye and G. Fang, "Motion Compensation Method Based on MFDF of Moving Target for UWB MIMO Through-Wall Radar System," in IEEE Geoscience and Remote Sensing Letters, doi: 10.1109/LGRS.2021.3116766.” (I’m not one the authors of this paper)
- The proposed method tends to remove drastic change in radar signal in a relatively short time by comparing the signal difference in the signal magnitude with time shift . However, how to determine the shift time is not clearly described. Is it a constant value or automatically determined by a certain algorithm? And what if signal compensation sector turns to zero in the time when it should be the peak or the trough of the respiration signal in the experiment? How would it influence the compensated signal and the measured respiration rate? The proposed method would be questionable theoretically if the above questions are not clearly explained.
- It could be better to prove the effectiveness by comparing the proposed method with other motion compensation method for radar signal in recent researches.
- The numerical analysis of experiment is not adequate to qualify the proposed motion compensation method in this article. Adding other metrics such as Signal Noise Ratio (SNR) of radar signals before and after procession can help proving the effectiveness of the method.
There are also several grammatical errors or statements that may lead to confusion as stated below:
- In the abstract, “ Theseresults confirm the possibility of using the proposed method in a non-statistic situation and effectiveness in estimating respiration rate reflecting human movement in monitoring vital signs using FMCW radar”
- In keywords, it would be clear to combine “moment; distortion; compensation” together and the same with “vital; respiration”
- In line 22, “As the number of vehicles rapidly increase, it is important to develop methods to prevent a large number of road accidents”.
- In formula (1) and (2), a bracket “)” is missing. Please check carefully.
- In formula (6), the meaning of is not explained.
6, In line 123, 127 and 128, the “Figure 1” should be “Figure 2” based on the context.
Reviewer 2 Report
The respiration rate extraction is not a new problem. In this manuscript, respiration rate extraction considering movement compensation is discussed. The application is interesting and practical. The experimental results show the validation of this application. However, there exist several problems need to be modified.
- The grammar and sentences need to be improved. And some errors should be revised. For example, in line 15, “this results”.
- The introduction is not complete. The traditional techniques of respiration rate extraction and movement compensation are not introduced.
- In line 82, “is the phase of previous signal”.
Does “This previous signal” point to reference signal?
How to choose the time internal of ?
- Describe the meaning of the figure 1a’ coordinate axes. How to get respiration of figure 1b.
- In real application, the driver may always move with the car. In this situation, how to compensate for the movement of body? Is the proposed method valid?
- The brackets are not balanced in formular (1) and (2).
Reviewer 3 Report
In this study, the author described a method to compensate for motion artifacts that could interfere with respiration rate measurement using an FMCW Radar.
The author demonstrated the various degree of performance improvements from the motion compensation technique in simulated driving environments.
Suggestions to the author include:
- The author described the working principle of an FMCW Radar but did not clearly specify in detail how different vital parameters are manifested (and how they are derived) in the FMCW Radar signal.
- The captions of many figures are vaguely written. Please consider rewriting them more clearly
- Many of the acronyms are not defined (such as IF)
- In Fig.2 where the author compares the acceleration measured by the accelerometer vs the moiton index, there does not seem to be obvious acceleration on body (where breathing motion happens) at ~time 10s. Please provide more explanation on why the radar detects the movement while the body sensor does not.
- The author needs to describe the components in the figures more clearly. In many figures, there are multiple sub-figures and the meaning of each sub-figure is not always clearly described.
- Please provide more analysis on how does the motion compensation work on various driving situations
- Please provide more information regarding the test results (such as bland-altman graph)
- The simulated driving condition seems to only consider the active motion from the driver, it is also worth discussing the motion from road bumpiness, etc.
- Please double-check typos and make improvements to English writings.